# A Tool to Design Bridging Oligos Used to Detect Pseudouridylation Sites on RNA after CMC Treatment

**DOI:** 10.3390/ncrna8050063

**Published:** 2022-09-23

**Authors:** Baptiste Bogard, Gilles Tellier, Claire Francastel, Florent Hubé

**Affiliations:** UMR7216 Épigénétique et Destin Cellulaire, CNRS, Université Paris Cité, F-75013 Paris, France

**Keywords:** CMC treatment, Oligonucleotide design, tool, Pseudouridylation, snoRNA

## Abstract

Pseudouridylation is one of the most abundant modifications found in RNAs. To identify the Pseudouridylation sites (Psi) in RNAs, several techniques have been developed, but the most common and robust is the CMC (N-cyclohexyl-N′-(2-morpholinoethyl)carbodiimide) treatment, which consists in the addition of an adduct on Psi that inhibits the reverse transcription. Here, we describe a turnkey method and a tool to design the bridging oligo (DBO), which is somewhat difficult to design. Finally, we propose a trouble-shooting guide to help users.

## 1. Introduction

In addition to the five canonical bases (ATGCU), about 150 chemically modified RNA nucleotides have been identified [1]. Pseudouridylation (Psi) can be added to RNA by on-autonomous enzymes, which are guided by small nucleolar RNAs (snoRNAs), or by specialized enzymes, such as autonomous pseudouridine synthases (Pus). Psi is among the most abundant modifications found in RNAs, from lincRNAs to tRNAs through mRNAs, and, of course rRNAs, the main targets of snoRNAs [2,3]. Psi was discovered over 60 years ago [4], and consists of the isomer form of uridine. As a consequence of isomerization, Psi increases both the rigidity of the backbone and the stability of Psi–A pairs compared to unmodified U–A [2].

Here, we describe a novel protocol aiming at the identification of Psi sites in RNAs. This protocol is an adaptation of the protocol by Zhang et al. [5]. We modified some steps to save time (the purification of the RNAs, for instance) and developed a tool, DBO (Design of Bridging Oligo), to design the bridging oligos (see Section 2). The main procedure is summarized in Figure 1.

The most common method currently used to identify Psi sites in RNAs is based on a CMC (N-cyclohexyl-N′-(2-morpholinoethyl)carbodiimide) treatment procedure [5,6,7]. CMC forms an adduct with the Psi base that stops the reverse-transcription reaction. Zhang et al. developed a method to quantitatively and qualitatively identify Psi sites in all kinds of RNA (mRNAs, lincRNAs, and rRNAs) [5]. As shown in Figure 1, they added a step to block RNA degradation that may interfere with the detection, using a ligation step with a 5′-end modified oligo (5AmMC6, 5′ Amino Modifier C6 modification). Herein, we propose the use of a specific tool, DBO, to design bridging oligos mimicking a double-stranded DNA to ligate the adaptor to the cDNA (see Figure 1). We solved some issues, such as the design of the bridging oligo, which is difficult, simplified the washes, and added a trouble-shooting section that may help users to navigate the procedure.

We used the SNORA70 Psi sites described in the 18S rRNA as a proof of principle (see [8]; snoRNABase, available online: https://www-snorna.biotoul.fr/plus.php?id=U70; accessed on 9 August 2022).

## 2. Results

### 2.1. Use of the DBO Tool

The tool is available at DBO. Available online: https://parisepigenetics.github.io/DBO/ (accessed on 9 August 2022).

The adaptor is ligated to the 3′-end of the cDNA of the target gene, which is reverse complementary to the RNA sequence. The size of the bridging oligo on adaptor side is usually 10 nucleotides but sometimes longer is better. The length may change depending on the experiment to increase specificity for instance. 

The RNA sequence is the sequence without Psi sites and the number of nucleotides selected depends of the Psi site position. 

The oligos to design are displayed directly with the sequence of the oligo to order. It is feasible to add as many oligos as needed using the table (add button) and export the table in a CSV file to ease ordering (Figure 2). 

### 2.2. PCR Results

As shown in Figure 3, the first PCR did not yield any PCR product; which is why we used a nested PCR to specifically amplify the PCR product. In some cases, you may have a PCR product already in the first PCR The PCR product was then cloned into a pGEM-T easy plasmid and sent to Sanger sequencing (Eurofins, Cologne, Germany).

### 2.3. Sequencing Results

As shown in Figure 4, the sequencing of the PCR product retrieved the plasmid (green), the adaptor oligo (yellow), and the expected part of the 18S rRNA (red), and we were able to identify the correct U Psi site described previously (see [8]; snoRNABase, available online: https://www-snorna.biotoul.fr/plus.php?id=U70; accessed on 9 August 2022).

## 3. Conclusions

This protocol was adapted from Zhang et al. [5]. It is more detailed and slightly faster, with a troubleshooting section and a tool called DBO to draw particularly difficult primers, such as bridging oligos.

Because we experienced many drawbacks during the setting of the protocol, we are now able to describe a very precise and detailed protocol that may be useful to any researchers who want to study Psi in total RNA or even in mRNA (not shown herein, to be published elsewhere). 

In addition to the DBO tool (DBO, available online: https://parisepigenetics.github.io/DBO/, accessed on 9 August 2022) that we propose here, there are two main differences from the original protocol proposed by Zhang et al. [5]. First, we cannot consider our protocol as being quantitative, as is the case in Zhang et al., mainly because a nested PCR is performed at the end, which creates PCR amplification biases. Second, the main drawback of both techniques is the primes efficiency to amplify the “hybrid” fragments in PCR reactions. Indeed, if primers do not work together (for instance, if the Tms are too different from each other), no PCR product is visualized. We overcame this problem with a nested PCR step, which adds a layer of specificity to the PCR amplification. As shown in Figure 3, no PCR product was observed after the PCR1, although 18S rRNA was abundant in the cells and the Psi sites were clearly identified, whereas it was overcome with a second step of nested PCR.

Although we have shown that this method is efficient for detecting already-known Psi sites in abundant and highly modified rRNAs, the question of the modification of low-abundance lncRNA or mRNAs remains challenging. We will provide evidence that this method is useful to tackle this question in a future manuscript.

Overall, this study presents a robust protocol that does not need to be implemented and a very useful tool, DBO, to design bridging oligos.

## 4. Materials and Methods

List of reagent

Tri-reagent (Sigma-Aldrich, St. Louis, MO, USA, cat. no. T9424)Chloroform (Sigma-Aldrich, cat. no. C2432)Ethanol absolute (VWR chemicals, cat. no. 64-17-5)Glycogen (Roche, cat. no. 10901393001)2-Propanol (isopropanol, Sigma-Aldrich, cat. no. 33539-M)Phenol:Chloroform:Iso-amyl alcohol (Sigma-Aldrich, cat. no. P1944)N-Cyclohexyl-N′-(2-morpholinoethyl)carbodiimide methyl-p-toluenesulfonate (CMCT, Sigma-Aldrich, cat. no. C106402)Tris base (Euromedex, Souffelweyersheim, France, cat. no. 200923)Hydrochloric acid (HCl, Sigma-Aldrich, cat no. 320331)Urea (Sigma-Aldrich, cat. no. U5378)EDTA, Tetrasodium Tetrahydrate Salt (Calbiochem, cat. no. 34103-M)Potassium acetate (KOAc, Sigma-Aldrich, cat. no. P1190)Potassium chloride (KCl, Sigma-Aldrich, cat. no. P9541)Sodium carbonate (Na_2_CO_3_, Sigma-Aldrich, cat. no. S2127)Murine RNase inhibitor (New England BioLabs, Ipswich, M, USA, cat. no. M0314L)T4 Polynucleotide kinase (PNK, New England BioLabs, cat. no. M0201L)T4 Polynucleotide Kinase Reaction Buffer, 10x (supplied with T4 Polynucleotide kinase)Adenosine 5′-Triphosphate (supplied with T4 RNA Ligase 1)Dimethyl sulfoxide (DMSO, Sigma-Aldrich, cat. no. D4540)T4 RNA Ligase 1 (New England BioLabs, cat. no. M0204S)T4 RNA Ligase Reaction Buffer, 10x (supplied with T4 RNA Ligase 1)Blocking oligo: 5′-[AmC6]ACCCA-3′ (Eurofins Genomics, Luxembourg)RevertAid H Minus Reverse Transcriptase (Thermo Scientific, Waltham, MA, USA, cat. no. EP0452)RevertAid H Minus Reverse Transcriptase Reaction Buffer, 5× (supplied with RevertAid H Minus Reverse Transcriptase)Random hexamer primer (Thermo Scientific, cat. no. SO142)Deoxynucleotide (dNTP) Solution Set (New England BioLabs, cat. no. N0446S)18s RT primer: 5′-ATCCGAGGGCCTCACTAAAC-3′RNase H, recombinant (New England BioLabs, cat. no. M0297L)Adaptor: 5′-TTTCTACTCCTTCAGTCCATGTCAGTGTCCTCGTGCTCCAGTCG-3′ (Eurofins Genomics)18s Bridging oligo: 5′-CTGAAGGAGTAGAAAGTACACACCGCCCGT[SpC3]-3′ (Eurofins Genomics)T4 DNA Ligase (New England BioLabs, cat. no. M0202S)T4 DNA Ligase Reaction Buffer, 10× (supplied with T4 DNA Ligase)OneTaq^®^ DNA Polymerase (New England BioLabs, cat. no. M0480L)OneTaq^®^ Standard Reaction Buffer, 5× (supplied with OneTaq^®^ DNA Polymerase)Adaptor primer: 5′-CGACTGGAGCACGAGGACACTGA-3′ (Eurofins Genomics)Adaptor nested primer: 5′-GGACACTGACATGGACTGAAGGAGTA-3′ (Eurofins Genomics)18s nested primer: 5′-ACCATCCAATCGGTAGTAGCG-3′ (Eurofins Genomics)Ethidium Bromide (BET, Euromedex, cat. no. EU0070)Acrylamide/Bis-acrylamide (19:1) (Sigma-Aldrich, cat. no. A3449)Ammonium persulfate (APS, Sigma-Aldrich, cat. no. A3678)N,N,N′,N′-Tetramethylethylenediamine (TEMED, Sigma-Aldrich, cat. no. T9281)Tris Borate EDTA (TBE) Buffer, 10× (Euromedex, cat. no. ET020-B)TriDye™ Ultra Low Range DNA Ladder (New England BioLabs, cat. no. N0558S)Gel Loading Dye, Purple (6×), no SDS (New England BioLabs, cat. no. B7025S)pGEM^®^-T Easy Vector System I (Promega, Madison, WI, USA, cat. no. A1360)NEB^®^ 5-alpha Competent *E. coli* (New England BioLabs, C2987)List of equipmentFilter units to sterilize buffer solutions (Thermo Scientific, Nalgene rapid flow filters)Thermoblock with shaking (Eppendorf Thermomixer comfort, Eppendorf, Montesson, France)Refrigerated centrifuge (Eppendorf, cat. no. 5415R)VortexLuria broth (LB) agar plates with ampicillin for bacterial selectionScalpelsVertical electrophoresis system (Hoefer Inc., Holliston, MA, USA, cat. no. SE400)Power supply unitsGel imaging system (Bio-Rad, Hercules, CA, USA, Gel Doc XR+)Reagent buffersTris-EDTA-Urea (TEU) buffer: Mix 50 mM Tris-HCl (pH 8.3), 4 mM EDTA and 7 M urea. Store at room temperatureCMC solution: 1M CMC prepared in TEU buffer. To be prepared extemporaneously.Reverse-reaction buffer: Mix 50 mM Na_2_CO_3_ and 2 mM EDTA. Adjust pH at 10.4. Store at room temperatureNon-denaturing polyacrylamide gel: Gel prepared with 10% acrylamide (19:1), 1× TBE in waterBET working solution: Prepare fresh, dilute stock solution with 1× TBE to use at 0.5 µg/mL

### 4.1. CMC Treatment

Five micrograms of total RNA (max volume 12 µL) were treated with CMC in TEU buffer for 16 h/18 h at 30 °C in 40 µL. A negative control without CMC was used in the reaction. Briefly, the reaction was composed of 12 µL of RNA (maximum volume) +24 µL TEU buffer +/− 4 µL CMC buffer, which formed the TEU buffer at 0.7×. After CMC treatment, RNA was purified by phenol/chloroform method to remove excess CMC. To reverse reaction, RNAs were incubated 6 h at 37 °C in 40 µL of reverse-reaction buffer. Another purification with phenol/chloroform method was applied and RNAs were resuspended in 6.5 µL H_2_O.

Suggestions for future work: The CMC buffer must be made extemporaneously (TEU can be kept at RT). The CMC should be resuspended in TEU buffer.Add glycogen (20 µg) during the phenol/chloroform procedure.If the target to be recovered is an mRNA, use 10 µg of mRNA instead of the 10 µg of total RNA.

### 4.2. RNA 5′ Phosphorylation

The 6.5 µL of RNAs were incubated with 0.5 µL RNase inhibitor, 1 µL of 10× T4 PNK buffer, 1 µL ATP, and 1 µL T4 PNK enzyme at 37 °C for 30 min. 

### 4.3. Ligation of the 5′ RNA Blocking Oligo

The 10 µL of RNA were supplemented with 1 µL of 10× T4 RNA ligase buffer, 1 µL of 100 µM blocking oligo (5′-[AmC6]ACCCA-3′), 1 µL of RNase inhibitor, 3 µL of DMSO, 2 µL H_2_O, and 1 µL of T4 RNA ligase I, and incubated at 16 °C for 16 h/20 h. The reaction was stopped by adding 1.2 µL of 200 mM EDTA. Final reaction volume was 21.2 µL.

### 4.4. Reverse Transcription

In total, 3 µL of previous ligation product were used, and 9 µL H_2_O, 2 µL of 1.5 µM RT primers (specific to the gene you want to identify the Psi site), and 1 µL of 10 mM dNTP were added. The mixture was heated at 65 °C during 2 min and placed immediately on ice, after which 4 µL of 5× RevertAid buffer, 0.5 µL RNase inhibitor, and 0.5 µL of RevertAid H Minus Reverse Transcriptase (200 U/µL) were added. The mixture was incubated for 20 min at 45 °C and the reaction was stopped by heating at 85 °C for 2 min. RNAs were degraded by adding 1 µL of RNaseH and incubated 20 min at 37 °C. RNaseH was inactivated by heating at 85 °C for 2 min.

### 4.5. Adaptor Ligation

As the T4 DNA ligase did not ligate single-stranded DNA, a bridging oligo is used to mimic a double strand (see Figure 1). To the 20 µL of cDNA, add 1 µL of 1.5 µM Adaptor, 1 µL of 1.5 µM bridging oligo (designed with the tool DBO presented herein), 4 µL of 10× DNA ligase buffer, 1 µL T4 DNA ligase, 1 µL ATP, 2.4 µL DMSO (6%), qsp at 40 µL H_2_O (9.6 µL). The ligation reaction was performed at 16 °C for 16 h/20 h and DNA ligase was inactivated by heating at 65 °C for 10 min.

Suggestions for future work: Adaptor, as almost all commercial oligos, is not phosphorylated in 5′-end. Treat the adaptor with T4 PNK before use, as previously described (Section 4.2).Use the DBO tool provided to design bridging oligos (see Figure 2).

### 4.6. PCR Amplification

To amplify the PCR products, 2 µL of the previous ligation reaction were used and amplification was performed as previously described [9]. 

Suggestions for future research: To observe a Psi site in 18S or 28S rRNA, apply a PCR of 15 cycles only. To view Psi site in mRNA, perform 35 cycles PCR.If the first PCR does not produce a PCR product or non-specific product, use nested PCR to amplify the PCR products more specifically (see Figure 1).

### 4.7. Sequencing

PCR product was cloned into pGEM-T easy plasmid and 3 minipreps were sent to Sanger sequencing (Eurofins) using T7 promoter oligo (one example is given in Figure 4).

### 4.8. Troubleshooting

Here, we provide a troubleshooting guide that can help users to overcome problematic issues. It consists of problems experienced during the setting up of the protocol, and the solutions found to resolve them. The guide is not exhaustive, but we would be happy to answer questions that readers may have.
No conversion/modification using CMCUse exclusively CMC solution prepared extemporaneously.CMC is resuspended in TEU buffer (insoluble in water).The final concentration of TEU buffer should be closed to 0.7× (from 0.5 to 1×). Adjust the volume of RNA accordingly.The conversion can be assessed by performing a PCR of the 18S using regular RT-PCR oligo on the RT products. If a product is obtained only in CMCT minus, this indicates that the conversion has worked (suggesting the blocking of the RT by the CMC worked).No RT productIf researchers use their own reverse transcriptase, they may need to decrease the incubation temperature and increase the reverse-transcription reaction to 30 min.No adaptor ligationDo not forget the 5′-end phosphorylation of the adaptor. Treat the adaptor as described in Section 4.2.To avoid ordering the incorrect bridging oligo, use the tool DBO supplied with this paper.The size of the bridging oligo can be increased; usually, 10 nt each side is enough, although, sometimes, a longer oligo is better.No PCR productThe easiest explanation is that the predicted Psi site does not correspond to the real site.Use a nested PCR to increase the specificity.To identify a Psi site on a target that is not an rRNA, increase the number of PCR cycles, or enrich the starting material in mRNA, for instance.Try to use touchdown PCR to enrich amplification templates.

## Figures and Tables

**Figure 1 ncrna-08-00063-f001:**
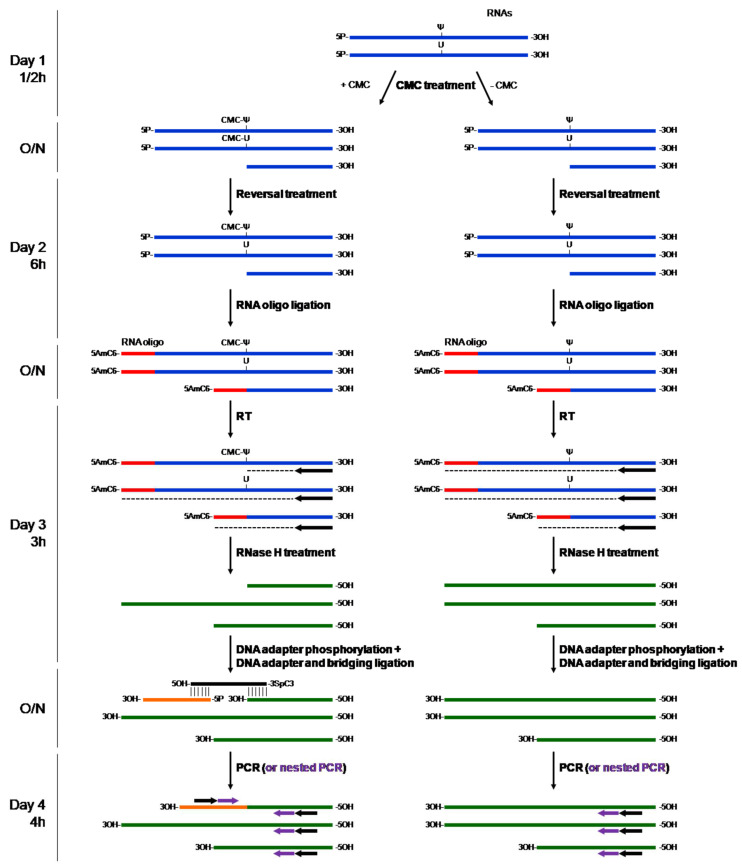
Principle of the CMC treatment followed by PCR. Total RNA was treated with either CMC (+CMC) or H_2_O (−CMC) for at least 16 h. Notably, random cleavage was produced during the treatment. Since an alkaline-reversal treatment was performed to remove the CMC-U, only the CMC-Ψ adducts remained. Next, a blocking RNA oligo was added at the 5′-end of the treated RNAs to prevent the potential fragmented RNAs from mimicking a RT-stop due to a CMC-Ψ adduct. After the reverse transcription (RT) and the RNase H treatment steps, a DNA adapter was added with the help of a bridging oligo, designed with DBO, which was half complementary to the 5′ end of the adapter and to the 3′ end of the RT-stop cDNA. Finally, only the generated fragment was amplified by PCR. If needed, a nested PCR can be performed to amplify the appropriate fragment more specifically. As an indication, the timing for each general step is estimated on the left. O/N, overnight; CMC, N-Cyclohexyl-N′-(2-morpholinoethyl)carbodiimide methyl-p-toluenesulfonate; Ψ, pseudouridine; 5AmC6, 5′ Amino-C6 linker; 3SpC3, 3′ Spacer C3.

**Figure 2 ncrna-08-00063-f002:**
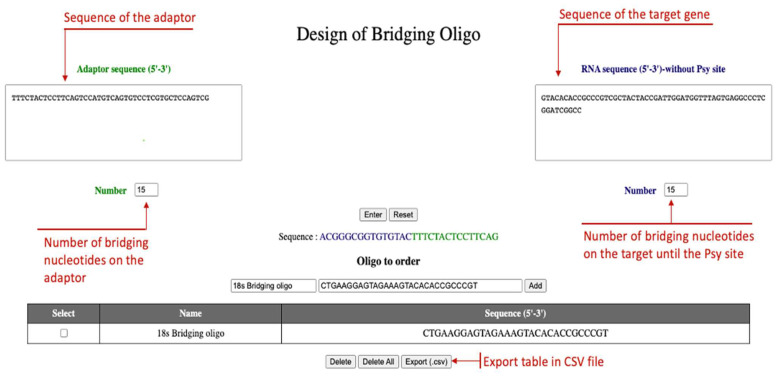
Use of DBO tool. The adaptor sequence is on the left box, the 18S RNA on the right box. The sequence of the 18S RNA started just after the Psi site to be tested. Pressing “Enter” will provide the intermediary hybrid oligo (before transformation) and the “Oligo to order” will give the oligo as it has to be ordered. By clicking the “Add” button, oligos are added to a list that is exportable to a CSV file to ease the ordering.

**Figure 3 ncrna-08-00063-f003:**
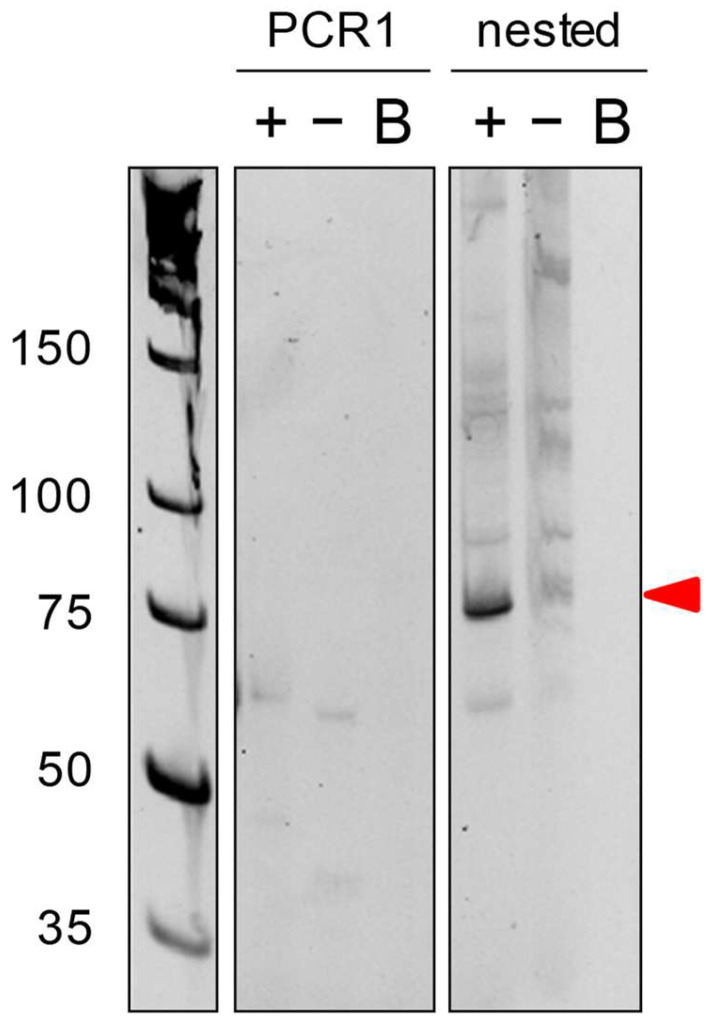
PCR result. PCR1 was performed using distal primers in the presence of CMC treatment (+), in absence of CMC treatment (−), or without cDNA (B). Since the PCR1 did not produce any amplicons, we performed nested PCR using nested primers. The amplicon of the correct size is shown by the red arrow.

**Figure 4 ncrna-08-00063-f004:**
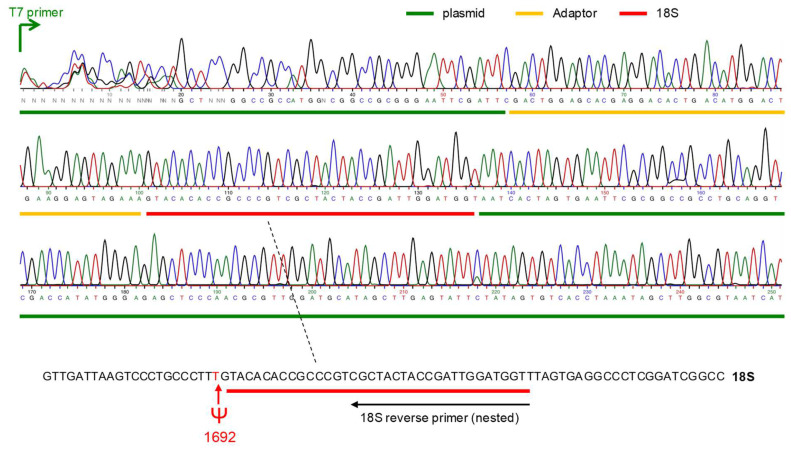
Sequencing result. The chromatogram of one plasmid sequencing result is shown here. The plasmid is in green, the adaptor in yellow, and the expected sequence of the 18S RNA in red. At the bottom, the Psi site identified in position 1692 is the site described in (see [8]; snoRNABase, available online: https://www-snorna.biotoul.fr/plus.php?id=U70; accessed on 9 August 2022).

## Data Availability

Not applicable.

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
