# Peer review of "A Tool to Design Bridging Oligos Used to Detect Pseudouridylation Sites on RNA after CMC Treatment"

_ncrna, 2022, doi:10.3390/ncrna8050063_

Round 1

Reviewer 1 Report

Mapping of post-transcriptionally modified ribonucleotides in cellular RNAs is an emerging challenge of modern molecular biology. Detection of pseudouridine (Psi) is based mostly on its specific and irreversible reaction with CMC and the fact that the resulting bulky Psi-CMC adduct arrests reverse transcriptase (RT). Recently, Zhang and his coworkers have reported an CMC treatment-reverse transcription-oligo ligation-PCR-based method for validation and quantification of predicted pseudouridines in rRNAs and mRNAs (RNA, 2019, 1218). In the current manuscript, Bogard et al propose a modified version of the Zhang method. Unfortunately, the alleged advantage (“slightly shorter”) of the new method over the published procedure of  Zhang is not evident to me, and description of the protocol is quite wag in the manuscript. Both methods rely on joining an adapter oligo to the 3’ end of cDNAs terminated before the CMC-modified test pseudouridineby using oligo-directed single-stranded DNA ligation (splint ligation). In the Zhang’s protocol, the sequence of the adapter oligo corresponds to a test RNA fragment located further upstream from the Psi site. Therefore, PCR amplification of the adapter-tagged cDNA products by using a tag-specific primer results in two DNA products, a longer and/or a shorter DNA, representing unmodified and pseudouridylated test RNA copies, respectively, and thereby revealing local pseudouridylation levels. The newly proposed approach of Bogard et al. is using an arbitrary adapter oligo, therefore it is unable to determine pseudouridylation efficiency. Moreover, in contrast to the work of Zhang et al., the current manuscript of Bogard et al. lacks experimental evidence demonstrating that their method would be able to validate the presence of pseudouridines in low abundance RNAs, such as mRNAs and lncRNAs.

In conclusion,  I think that the proposed pseudouridylation mapping protocol does not contain significant new innovation and enough advancement which could justify its publication.

Finally, the authors also developed an informatic tool to design bridging oligos. Designing appropriate guide/bridging oligos for DNA splint ligation should not cause problem even for undergraduate students, but unfortunately, as I know from my own experience, this is not the case. So, this tool, through saving money, time and annoyance in the lab,  might be be useful for the community, and it should be available for the public.

Minor points

Abstract, first sentence.  “Pseudouridylation” a biochemical reaction and its product (pseudouridine, Psi) are mixed up.

Page 1. Parag. 2. I am not aware of any miRNA carrying documented Psi.

Author Response

See the Word document that contains a figure...

Reviewer 2 Report

Comments to Authors

The manuscript entitled “A tool to Design Bridging Oligos used to detect pseudouridylation sites on RNA after CMC treatment” by Baptiste Bogard et al., presents an improved protocol to detect pseudouridylation sites on RNA after CMC treatment and a novel tool for the design the Bridging oligos used in it. In my view, the article presents a good development, is interesting and provides information on a protocol and tool that could be very useful for researchers who wish to work on RNA characterization. So, it would be very useful for biochemical and bioinformatic researcher, and the scientific community in general. Therefore, it could be Accepted in the present form. However, I have three considerations which are outlined below.

- In the list of agents used (Materials and Methods) there are some that seem to be with another letter or format (lines 10 to 167).

- I think section 4.8. Trouble shooting, could be modified to make it more compatible with the Materials and Methods format. Perhaps it could be transformed into a table with two columns (Problems; Solutions). At the same time, write a short paragraph indicating that they will present some of the problems that researchers could face and the possible solutions proposed by the authors.

- When the authors refer to the site where they sequenced or where they ordered the oligos, it appears in different ways throughout the manuscript; it would be better to homogenize it. In some places it is indicated (Eurofins Genomics), in others it is indicated (Eurofins) and in others (Eurofins, Cologne, Germany).

Author Response

The manuscript entitled “A tool to Design Bridging Oligos used to detect pseudouridylation sites on RNA after CMC treatment” by Baptiste Bogard et al., presents an improved protocol to detect pseudouridylation sites on RNA after CMC treatment and a novel tool for the design the Bridging oligos used in it. In my view, the article presents a good development, is interesting and provides information on a protocol and tool that could be very useful for researchers who wish to work on RNA characterization. So, it would be very useful for biochemical and bioinformatic researcher, and the scientific community in general. Therefore, it could be Accepted in the present form. However, I have three considerations which are outlined below.

We thank Reviewer 2 for his comments.

- In the list of agents used (Materials and Methods) there are some that seem to be with another letter or format (lines 10 to 167).

We corrected this problem of format (coming from the conversion of the submitted manuscript to the version proposed to reviewers).

- I think section 4.8. Trouble shooting could be modified to make it more compatible with the Materials and Methods format. Perhaps it could be transformed into a table with two columns (Problems; Solutions). At the same time, write a short paragraph indicating that they will present some of the problems that researchers could face and the possible solutions proposed by the authors.

We are not sure to understand what is asked here as it is already a table with two columns. But we wrote a short paragraph as proposed. We added this paragraph:

Here we provide a trouble shooting guide that can help users to overcome problematic issues. It consists of problems experienced during the setting up of the protocol, and the solutions found to resolve them. This is not exhaustive but we would be happy to answer questions that the readers may have and we will try to help.

- When the authors refer to the site where they sequenced or where they ordered the oligos, it appears in different ways throughout the manuscript; it would be better to homogenize it. In some places it is indicated (Eurofins Genomics), in others it is indicated (Eurofins) and in others (Eurofins, Cologne, Germany).

This has been corrected, thanks to the reviewer.

Reviewer 3 Report

The authors have created a novel tool for designing bridging oligos used to detect pseudouridylation sites on RNA after CMC treatment. The method can be very useful for epi-transcriptomic studies and also as a base to design new software for other RNA modifications.

The work can be accepted for publication with minor review. 

1) The source code for the method should be made available to the scientific community as open source.

2) Slight finishing on english can be done.

3) Section like the method shortcomings and future prespectives and applications can be added.

Author Response

The authors have created a novel tool for designing bridging oligos used to detect pseudouridylation sites on RNA after CMC treatment. The method can be very useful for epi-transcriptomic studies and also as a base to design new software for other RNA modifications.

The work can be accepted for publication with minor review. 

We thank Reviewer 3 for such consideration.

1) The source code for the method should be made available to the scientific community as open source.

The source code will be available as soon as the manuscript is accepted for publication.

2) Slight finishing on english can be done.

We indeed proofread the manuscript according to the reviewer’s suggestion.

3) Section like the method shortcomings and future perspectives and applications can be added.

We added a paragraph in the conclusion:

In addition to the tool DBO (https://parisepigenetics.github.io/DBO/) that we propose here, there are two main differences with the original protocol proposed by Zhang et al. (5). First, we cannot consider our protocol as being quantitative as it is the case in Zhang et al., mainly because a nested PCR is performed at the end, which brings PCR amplification biases. Second, the main drawback of both techniques is the primers efficiency in PCR reactions to amplify the "hybrid" fragments. Indeed, if primers did not work together (Tm were too much different for instance), no PCR product will be visualized. We overcame this problem with a nested PCR step, which adds a layer of specificity in the PCR amplification. As seen in Figure 3, no PCR product was observed after PCR1 although 18S rRNA is abundant in the cell and Psi sites well-identified, whereas it was overcame with a second step of nested PCR.

Although we have shown that this method is efficient in detecting already known Psi sites in abundant and highly modified rRNAs, the question of the modification of low abundance lncRNA or mRNAs remains challenging to address. We will provide evidence that this method will be useful to tackle this question in a manuscript to be published elsewhere.

Round 2

Reviewer 1 Report

I support publication of the revised manuscript, although it still needs English correction.